# A Dual-Inlet Pump with a Simple Valves System

**DOI:** 10.3390/mi14091733

**Published:** 2023-09-04

**Authors:** Le Wang, Junming Liu, Xin Wang, Si Chen

**Affiliations:** 1School of Intelligent Manufacturing and Elevator Mechanics, Huzhou Vocational & Technical College, Huzhou 313099, China; 2School of Mechanical Engineering, Guizhou University, Guiyang 550025, China; 3College of Mechanical and Electrical Engineering, Wenzhou University, Wenzhou 325035, China; 22451439030@stu.wzu.edu.cn (X.W.); 20200316@wzu.edu.cn (S.C.)

**Keywords:** piezoelectric pump, output flow performance, cantilever valve, stiffness optimization, microfluidic

## Abstract

To ameliorate the deficient output flow performance of the piezoelectric pumps with cantilever valves, a dual-inlet pump with a simple valves system is proposed. On the basis of explaining the structure design of the prototype, the working principle of the prototype is explained, and the output flow is theoretically analyzed. Then, the manufacturing process of the prototype is introduced. The general operating frequency of the pump was obtained by combining the finite element analysis of the piezoelectric actuator under dry and wet modes with the mechanical vibration test, and a series of measured flow rates were compared and improved through valve stiffness optimization and pump chamber height adjustment in the subsequent control experiment. The proposed piezoelectric pump achieves a maximum flow rate of 33.18 mL/min at a 180 V_p-p_ voltage with the driving frequency of 100 Hz, which may bring new inspiration for the application of small intelligent pumps in the field of microfluidics.

## 1. Introduction

Considering the advantages of small size, lightweight, fast response, low power consumption, and stable driving ability [1,2,3,4], piezoelectric pumps have played an important role in the fields of avionics [3,5,6], micro-robotics [7,8,9], atomization devices [10,11,12], and bioengineering [13,14,15]. Based on whether valve structures are used inside the pump body, piezoelectric pumps can be divided into valved pumps and valveless pumps, which have been studied to various degrees in the past few decades.

A typical valveless piezoelectric pump utilizes the resistance difference between positive and reverse flow to ensure unidirectional flow of liquids [16], and the simple manufacturing makes it easy to realize miniaturization. However, the phenomenon of backflow leads to a low work efficiency [17,18], which weakens the practical application of the valveless pumps. To improve this constraint, Yang et al. presented a valveless micropump with double piezoelectric vibrators in series connection [19]. He et al. designed a special pump on the basis of the Coanda effect using a bluff-body structure to increase the net flow [20]. Bao et al. proposed a piezoelectric pump with a built-in compliant structure that combined the properties of both valved pumps and valveless pumps [21]. Besides, a new-type pump with arc-shaped inlet and outlet tubes used for navigation equipment has also been mentioned [22]. Nevertheless, these works generally have deficiencies of high driving conditions and a low flow volume. The development of high-output valved pumps is still the mainstream of research.

Among the various branches of valved piezoelectric pumps, passive check valves are the most widely studied object. In order to achieve high back pressure and flow output, many pumps of wheeled valves [23,24,25] and umbrella valves with a wide flow area and low opening pressure have been developed [26,27,28]. However, the problems of high processing difficulty and the bulky shape of the two forms are obvious. Other forms, such as the flexible film valve [29], cymbal-shaped slotted valve [30], and venous imitation valve, have also been presented [31,32]. Moreover, the poor precision of the valve body and the insufficient cut-off are their common shortcomings. In contrast, the features of simple structure, low cost, and being conducive to miniaturization endow the cantilever valve a certain research potential in narrow spaces, such as micro-fuel-cells and gas–liquid conveying devices [33,34,35]. Through tests on cantilever beams of different lengths, Kan et al. proposed a piezoelectric pump with a flow rate of 3.5 mL/min [36]. Ma et al. designed and improved a micropump with a single-side-driven diaphragm for the circulation liquid cooling system of laptops [37,38]. Woo et al. developed a reciprocating pump with a reed valve made of PET and discussed the relationship between the shape of the valve and the pump performance [39]. The researchers at Jilin University compared the performances of cantilever valve pumps with different chamber numbers, in which the four-chamber series pump can obtain the maximum flow rate of 110 mL/min at a 170 V_p-p_ voltage of 450 Hz [40]. Then, they proposed a micropump with a length and width of only 11 mm in the same year [41], which provided inspiration for the design of microelectronic devices in the medical field. In addition, a micropump with both piezoelectric resonance and fluid inertia effects was mentioned [42]. It should be noted that small flow rates caused by an insufficient cut-off are apparent defects of cantilever valve pumps. Therefore, further exploration is needed to optimize the output flow performance of the piezoelectric pump by improving the valve functions as much as possible on the premise of ensuring the compact and simple structure of the pump body.

A piezoelectric pump with a dual-inlet structure using a simple cantilever valves system is presented in this work, which can attain a flow rate of 33.18 mL/min under a driving voltage of 180 V_p-p_ and an excitation frequency of 100 Hz. First, the internal structure and fabrication process of the pump are elaborated. Then, the finite element simulations under dry and wet modes of the piezoelectric actuator with thickness-stretching-vibration mode are conducted, and its resonance frequency is verified through laser vibration measurement. Finally, the prototype is assembled, the output flow rates of the pump are tested, and the output flow performance is enhanced by optimizing the stiffness of the valve system and the height of the pump chamber.

## 2. Design and Operating Principle

### 2.1. Construction of the Pump

As illustrated in Figure 1, the main body of the pump is a sandwich structure composed of the pump cover, the upper pump body, and the lower pump body. These three parts are fixed to four pairs of bolts and nuts. As the core component of the pump, the upper pump body provides an annular groove which allows the seal rings and the piezoelectric actuator to be embedded in it. The two symmetrical inlet valve grooves on the surface and the outlet valve groove on the bottom are compatible with the valve slices, and the positions of the three groove holes correspond to the flow holes of the lower pump body, in sequence. The three-valve system composed of inlet and outlet valves ensures the steady unidirectional flow of liquid when the piezoelectric actuator is working.

Figure 2 provides the main structural dimensions in the top and section view of the upper pump body, and their parameters are listed in Table 1. The annular groove on the top has a diameter of 35 mm and a depth of 1.2 mm. These sizes make the metal substrate of the piezoelectric actuator and its sealing rings on both sides fit precisely. Based on empirical values, the initial height of the pump chamber is 0.6 mm, and the depth of all valve grooves is 0.8 mm to maintain a certain opening displacement of the valves. Further, the inlet slot holes with a diameter of 5 mm and the flow holes with a diameter of 3 mm create a slope structure above the flow channel, which is beneficial for liquid suction. An outer diameter of 48 mm and a total height of 22 mm make the pump construction relatively simple and compact.

### 2.2. Working Principle of the Pump

As the direct driving element of the piezoelectric pump, the piezoelectric actuator converts electrical energy into mechanical energy by virtue of the inverse piezoelectric effect, which provides the power source for the pump; thus, the liquid is prompted to carry out the directional continuous flow under the effect of the check valves system. Figure 3 shows a demonstration of the working process of the piezoelectric pump, which mainly contains the following three steps:Step 1.Voltage excitation is applied to the surface of the piezoelectric actuator to make it bend and vibrate under the action of an external electric field. When the piezoelectric actuator bends upward, the volume of the pump chamber increases and the internal pressure decreases. Once the external pressure is greater than the opening pressure of the inlet valves, the inlet valves open and the outlet valve closes. Then, the liquid is absorbed into the pump.Step 2.After bending up to the extreme point, the piezoelectric actuator gradually returns to the equilibrium state. The fluid pressure is balanced with the external pressure at this time, and the inlet and outlet valves are closed. Then, the liquid is retained in the pump.Step 3.Next, the piezoelectric actuator bends downwards, the volume of the pump chamber decreases, and the liquid pressure increases. Once the liquid pressure is greater than the opening pressure of the outlet valve, the outlet valve opens and the inlet valves close. Then, the liquid is discharged from the pump.

Thereby, the piezoelectric actuator vibrates reciprocally in response to the periodic voltage. The liquid drainage and suction process is constantly repeated to reach the working purpose of the piezoelectric pump.

### 2.3. Theoretical Analysis of the Pump

For most piezoelectric pumps, the key influence factor of the ultimate output flow is the volume change quantity per unit working period. When the piezoelectric actuator is at the extreme positions of vibration, the maximum value of the volume change quantity, $\Delta V_{\max}$, in the pump chamber can be regarded as the volume of a fusiform body composed of two curved surfaces, both above and below. By leading into ω(φ), which is the deformation curve equation of the piezoelectric actuator, $\Delta V_{\max}$ can be integrated as follows [43]:(1)$$\Delta V_{\max}=\int_{0}^{\varphi }\omega \left(\varphi \right)\varphi d\varphi$$

The vibration deformation at the edge of the piezoelectric actuator pressed by the pump cover and the sealing rings is so weak that it can be ignored, so Equation (1) is taken as the actual diameter. Here, the relationship between the maximum output flow, $Q_{\max}$, and $\Delta V_{\max}$ can be written as:(2)$$Q_{\max}=\Delta V_{\max}c_{v}f$$
where $c_{v}$ is the response coefficient of the valve and f is the driving frequency. Neglecting the fluid loss of the narrow gap between the cantilever valves and the flow channels, the Bernoulli equation can be transformed into the following simplification under the condition that the inlet and outlet of the pump are equal in height [44]:(3)$$p_{1}+\frac{1}{2}\rho v_{1}{}^{2}=p_{2}+\frac{1}{2}\rho v_{2}{}^{2}$$
where ρ is the fluid density, $p_{1}$ and $p_{2}$ are the pressures at the pump chamber and the outlet, respectively, while $v_{1}$ and $v_{2}$ are the flow velocities at the pump chamber and the outlet, respectively. With a slight modification of Equation (3), we obtain:(4)$$\Delta P=\frac{1}{2}\rho \left(v_{1}{}^{2}-v_{2}{}^{2}\right)$$
where ΔP is the pressure variation of the pump. By contacting the relationship between output flow and fluid velocity, Equation (4) can be transformed into:(5)$$\Delta P_{\max}=\frac{1}{2}\rho \frac{Q_{\max}}{k_{d}}\left(\frac{1}{A_{1}{}^{2}}-\frac{1}{A_{2}{}^{2}}\right)$$
where $k_{d}$ is the flow coefficient, while $A_{1}$ and $A_{2}$ represent the cross-sectional area of the pump chamber and the outlet area, respectively. According to the formulas above, the limit values of the flow rate and pressure of the pump can be calculated, and the theoretical data of the output performances can be acquired.

## 3. Simulation and Verification

### 3.1. Fabrication Process

According to the structural design of the pump, the process of the prototype was planned as shown in Figure 4. Considering the physical properties and cost of the material, transparent UV-curable resin was selected as the raw material to produce the pump components, and a 0.5 mm-thick brass belt was used for the cantilever valves. Through 3D printing (J850 DAP, SoonSer, Shenzhen, China) and laser cutting (0606D, BKIaser, Hangzhou, China), the corresponding parts can be separately produced. Next, 30 min of ultrasonic cleaning was performed to remove the impurities. Then, the cleaned valve slices were glued to the valve slots of the upper pump body by a strong adhesive (BC-8528, Beichuang Rubber Industry, Jiangmen, China) and dried in a vacuum environment of 60 °C for 2 h; in this way, the effectiveness of the produced valves can be ensured.

For the manufacture of the piezoelectric actuator, a disc-shaped brass substrate with a diameter of 35 mm and a piezoelectric transducer with a diameter of 25 mm were bonded by epoxy resin and cured at 70 °C for 3 h, both of which were 0.2 mm in thickness. Furthermore, a pair of 0.5 mm-wide sealing rings, custom-made by nitrile butadiene rubber (NBR), were respectively placed on the upper and lower surfaces of the piezoelectric actuator, which acted as the elastic buffer and liquid isolation. Then, the whole pump prototype can finally be assembled.

### 3.2. Finite Element Analysis of the Piezoelectric Actuator

To preliminarily obtain the approximate operating frequency of the piezoelectric pump, modal analysis and harmonic response analysis of the piezoelectric actuator are needed first. Hence, the finite element software ANSYS Workbench (2021b) was used to complete the relevant simulation and analysis. To simplify the simulation steps, the pump body and other geometric structures were omitted. The peripheral fixed constraints and loads were added to the piezoelectric actuator itself. Here, the liquid was water, the piezoelectric transducer was PZT-5H, and the metal substrate was brass. The specific material performance parameters are shown in Table 2.

Since the water flow in contact with the bottom of the piezoelectric actuator will have an influence on its vibration to some extent, the vibration modes and resonance frequencies of the piezoelectric actuator under dry and wet modes were compared by using the acoustic module. Figure 5 shows the first-order modal shapes of the piezoelectric actuator in both dry and wet modes and their typical amplitude–frequency characteristic curves. It is evident that their vibration modes resemble each other, whereby the resonance frequencies of the two modes were 102.86 Hz and 93.43 Hz. Vibration was suppressed due to the damping effect of water when the piezoelectric actuator was in the wet mode, causing its lower frequency and smaller peak vibration speed.

### 3.3. Test of Vibration Characteristics

After the finite element simulation, the actual mechanical vibration characteristics of the piezoelectric actuator were measured. Here, the frequency response characteristics were tested using a laser Doppler scanning vibrometer system (PSV-300 F-B, Polytec GmbH, Waldbrunn, Germany). The full set of test equipment shown in Figure 6a mainly includes the power amplifier, the control system, the monitor, the laser head, and the vice holding the pump prototype. The surface of the piezoelectric actuator was placed nearly parallel to the lens of the laser head by adjusting the vertical tilt angle of the pump.

The test result in Figure 6b shows that the resonance frequency under the out-of-plane vibration mode along the thickness direction of the PZT transducer was 98.86 Hz, which had an error of 3.42 Hz (3.34%) with the frequency phase in the dry mode obtained by the above simulation. Taking the material property differences, the settings of the boundary conditions, and the omission of electrodes and bonding layers in the physical model into account, this discrepancy was acceptable [45].

Observing the mode shape of the piezoelectric actuator under the driving frequency, that the surface of the PZT transducer performed both positive extension and negative contraction in the out-of-plane expansion mode. Although the vibration of the outer edge was slightly abnormal due to the fixed point and bolt stress, it was basically consistent with the finite element simulation phenomena. It can be summarized that through the mechanical vibration test, the working frequency of the piezoelectric actuator was confirmed, and the operating scheme design of the piezoelectric pump was reasonable.

## 4. Experiments and Summaries

### 4.1. Experimental Conditions

After the finite element simulation and the laser vibration test, a series of experiments were carried out on the output flow performance of the pump prototype. The testing devices and required instruments are depicted in Figure 7. The signal generator (DG2052, RIGOL, Wuhan, China) output an adjustable initial voltage signal between 0 and 10 V_p-p_, which can be enlarged 20 times by means of the power amplifier and the transformer (FPA1000, FeelElec, Zhengzhou, China). For the sake of reasonably controlling the voltage range, the oscilloscope (SDS1202X-C, Siglent, Shenzhen, China) was used to obtain real-time supervision of the voltage value. The pump prototype was clamped by a minor plain vice, and the three through-holes on its bottom were fixed to the plastic hoses. The ends of the two inlet pipes extended into the beakers located on both sides of the plain vice, which can be seen as the water source. The end of the outlet pipe was inserted into the measuring cylinder, and the flow rate of the pump was calculated by observing the scale of the water column rising per unit time.

### 4.2. Working Frequency of the Prototype

Figure 8a presents a photograph of the experimental site. The plain vice and the beakers were placed on cushions of different heights to keep the inlets of the pump prototype flat with the water surface in the beakers. A certain scale of water was injected into the measuring cylinder in advance to avoid the visual error caused by its base. The sine wave signal was chosen. Here, the excitation voltage was set to 150 V_p-p_ to obtain clear experimental effects, and the driving frequency was gradually adjusted from 40 Hz to 120 Hz at intervals of 5 Hz. From this, the observation points corresponding to the flow rate values of each frequency could be marked and fitted to form the curve described in Figure 8b.

In the diagram, the flow rate of the pump shows a trend of increasing first and then declining with the increasing frequency. The maximum value occurred at 100 Hz, and it approached the resonance frequencies of the piezoelectric actuator, as seen from the finite element simulation and the laser vibration measurement test results. When the frequency exceeded 100 Hz, the change in the pump chamber volume per unit time decreased due to the weakening of the piezoelectric actuator surface amplitude, leading to the decline of the flow rate. Therefore, 100 Hz was selected as the optimal driving frequency.

### 4.3. Valve Stiffness

Stiffness is a significant factor that directly affects the opening and closing characteristics of cantilever valves, and the optimization of valve stiffness had a positive effect on improving the output flow rate of the pump prototype. To describe the relationship between the parameters of the valve plate and its stiffness, the central junction was simplified as a rectangular model, as shown in Figure 9, where *b* stands for the length, *L* is the width, and *t* is the thickness. Then, the equivalent stiffness, *k_eff_,* can be expressed as [46]:(6)$$k_{eff}=\frac{ELt^{3}}{6b^{3}}$$
where *k_eff_* is proportional to *L* and *t*, and in reverse proportion to *b*. It is clear that appropriately reducing *L* and *t* or increasing *b* can reduce the stiffness of the individual valve plate, which may enhance the opening and closing performance of the valve system. In this experiment, the other parameters of the valves remained unchanged, and the stiffness was regulated by alternating the vertical distance, *L,* of the central connecting part.

For this purpose, the original valve system with *L* of 2 mm was supplemented with two groups of valves with *L* of 1.5 mm and 1 mm. To observe the relatively obvious experimental phenomena without causing the piezoelectric actuator to breakdown, we set the voltage to range from 120 V_p-p_ to 180 V_p-p_ at intervals of 10 V_p-p_ and set the driving frequency as the optimal value of 100 Hz. Then, Figure 10a can be plotted by sequentially investigating the variation rules of flow rate and voltage under different stiffness valve systems. The flow rate increased linearly as the voltage increased. The valve system with *L* of 1 mm had the best effect, followed by the valve system with *L* of 1.5 mm, and both were superior to the original valve system.

Figure 10b reveals the flow rate change of the pump prototype using the valve system with *L* of 1 mm within 60 s at the voltage of 180 V_p-p_, and the water volume of the measuring cylinder in the image increased from the pre-set 10 mL to nearly 42 mL. This phenomenon confirms that the cut-off performance of the cantilever valve was enhanced by optimizing its stiffness; thus, the output flow rate of the pump prototype was improved using this method.

### 4.4. Chamber Height

As well as the comparative analysis of the valve performance, optimizing the structural parameters of the pump body is also necessary. The height of the pump chamber, *h,* is crucial for indicating the volume change of the chamber, which means reasonably setting *h* is of value to improve the flow output performance. In light of the original pump prototype with *h* of 1 mm, similar types with *h* of 0.8 mm, 0.6 mm, and 0.4 mm were successively proposed. The valve system with *L* of 1 mm was adopted and the control experiment was assigned a voltage of 180 V_p-p_ at 100 Hz. The comparison is shown in Figure 11. The flow rate of each type followed a trend of increasing first and then decreasing with the increase of the driving voltage at different chamber heights. There was a negative correlation between the flow rate and the height of the pump chamber, and the maximum flow rate gained 33.18 mL/min when *h* was 0.4 mm. The aim of flow output performance improvement has been accomplished by optimizing the essential design parameter of chamber height.

### 4.5. Viscosity of Fluid 

After optimizing the valve stiffness and pump structural parameters, the influence of the inherent properties of liquids on the output flow performance of the pumps cannot be ignored. It is necessary to explore the viscosity of the fluid, which is a key factor causing the loss of flow [47]. Ethanol was selected as the experimental object, then each type of solution was prepared and tested sequentially at a voltage of 180 V_p-p_, as shown in Table 3. The flow rate changes of the ethanol solutions at different concentrations and the viscosity coefficients of different concentrations at room temperature are shown in Figure 12.

The flow rate trend of ethanol solution is similar to that of the pumped water medium. The 10% ethanol solution had a maximum flow rate of 30.7 mL/min at 95 Hz, the 20% ethanol solution had a maximum flow rate of 25.68 mL/min at 90 Hz, and the 30% ethanol solution had a maximum flow rate of 20.74 mL/min at 85 Hz. With the increase of viscosity, the maximum output flow rate decreased, and its corresponding driving frequency also slightly decreased. This is because increasing the viscosity provides additional mass and damping of the liquid, resulting in a lower natural frequency and a stronger damping effect of the piezoelectric pump. These changes weaken the efficiency of liquid transportation, and lead to a decrease of the flow rate. Thus, the appropriate liquid should be selected as the conveying medium according to the expected flow rate of the piezoelectric pump to achieve the most effective driving level in specific working situations.

Table 3 lists some typical piezoelectric pumps with cantilever valves, and they are compared with this work in terms of the output flow performance and driving conditions. The optimal operating frequencies of each prototype are similar, and the pump proposed in this work is slightly lower. While the voltage applied in this work was relatively high to achieve the ideal experimental results, the maximum output flow rate is evidently higher than that mentioned in other studies. Therefore, using the optimized cantilever valves system is beneficial to increase the pump flow.

## 5. Conclusions

To make progress on the low output flow of piezoelectric pumps with cantilever valves, a new type of pump with double inlets was proposed. On the premise of establishing the pump structure and working principle, relevant simulation and mechanical vibration characteristics tests were conducted, and comparative analysis and optimization were discussed for the output flow performances of different prototypes. The main conclusions are as follows:(1)In accordance with the finite element simulation results of the piezoelectric actuator under dry and wet modes, the amplitude frequency response characteristics displayed through laser scanning tests were analyzed. The first-order out-of-plane stretching vibration mode in the low frequency range confirmed the working mode of the pump; thus, the preliminary operating frequency was mastered.(2)By adjusting the physical characteristics of the cantilever valves and the structural parameter of the pump chamber, the improvement of the output flow performance was feasible. Reducing the stiffness of the valve helped to make its opening and closing responses better, and the liquid flowing through the pump chamber became more concentrated with a lower height.(3)The proposed piezoelectric pump prototype has a compact and simple configuration of a Φ 48 mm × 22 mm external size, with a maximum flow rate of 33.18 mL/min under a 180 V_p-p_ voltage at a driving frequency of 100 Hz. It may have potential application in places of microfluidics operation, such as liquid medicine transportation and agricultural drip irrigation.

## Figures and Tables

**Figure 1 micromachines-14-01733-f001:**
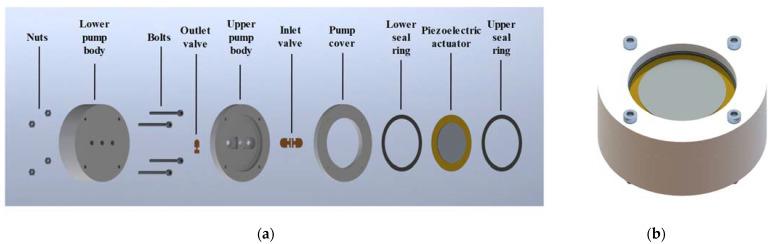
Structure of the piezoelectric pump, (**a**) 3D explosion structure of the prototype, and (**b**) overall appearance of the prototype.

**Figure 2 micromachines-14-01733-f002:**
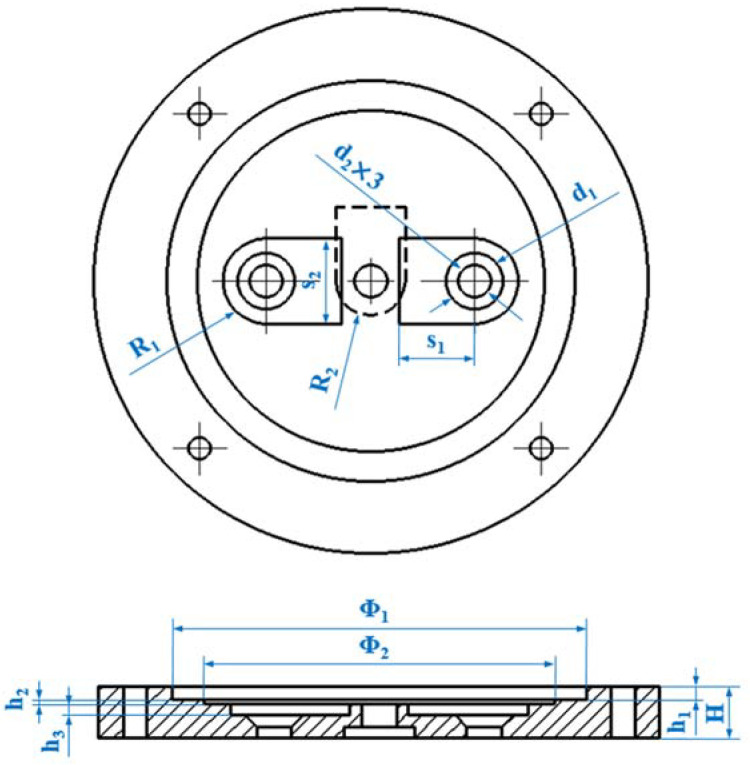
Main dimensions of the top and section views in the upper pump body.

**Figure 3 micromachines-14-01733-f003:**
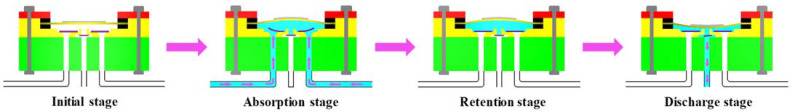
The four working stages of the piezoelectric pump.

**Figure 4 micromachines-14-01733-f004:**
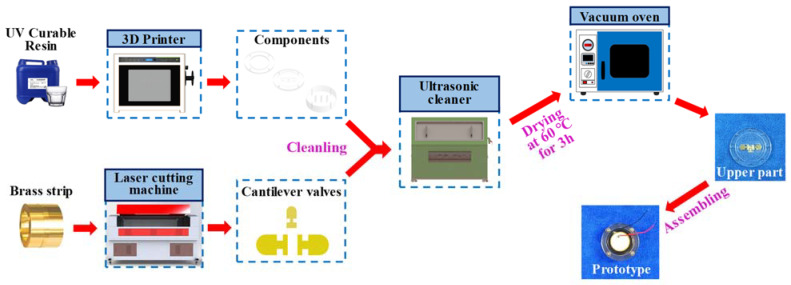
Manufacturing process of the pump prototype.

**Figure 5 micromachines-14-01733-f005:**
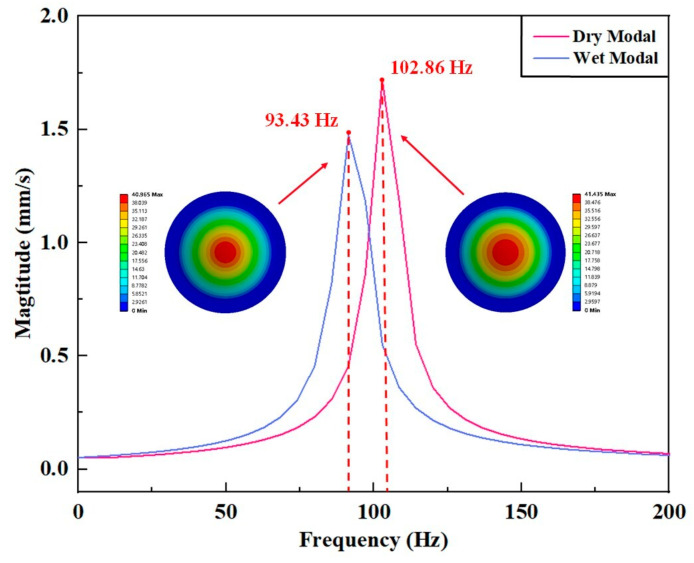
Comparison of dry and wet modes of the piezoelectric actuator.

**Figure 6 micromachines-14-01733-f006:**
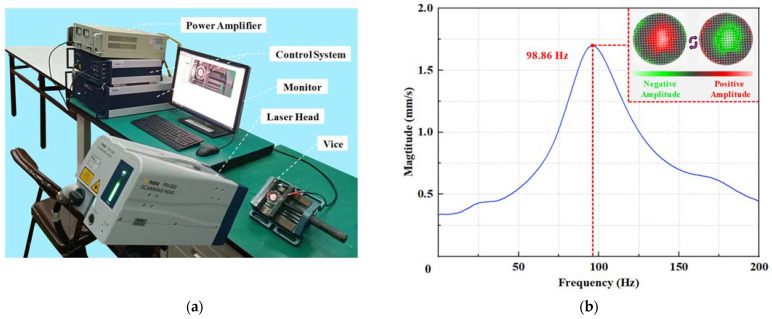
Mechanical vibration performance test of the piezoelectric actuator: (**a**) laser vibration measurement platform and (**b**) frequency response characteristics of the piezoelectric actuator.

**Figure 7 micromachines-14-01733-f007:**
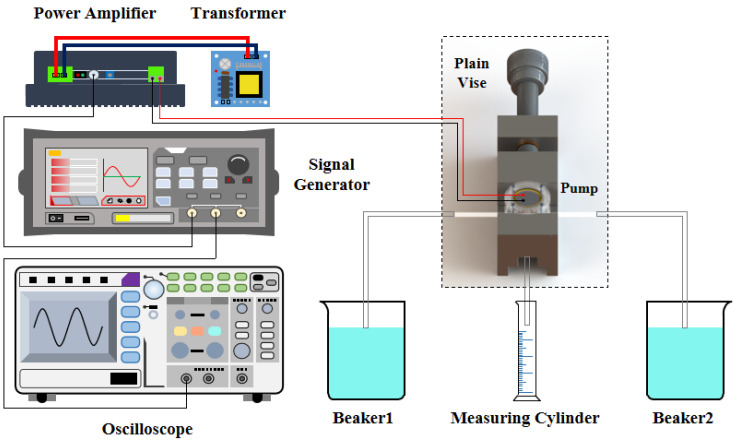
Arrangement of the output flow performance experiments of the pump prototype.

**Figure 8 micromachines-14-01733-f008:**
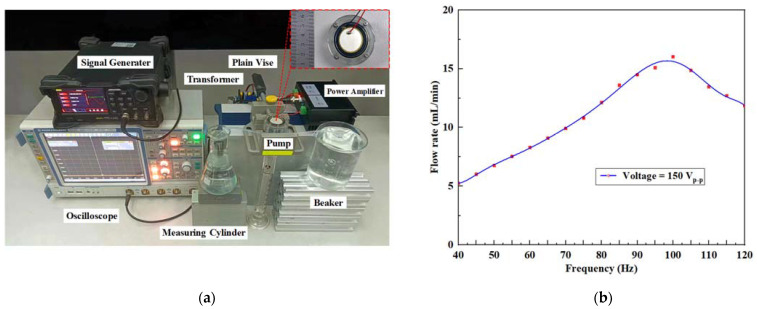
Determination of the optimal driving frequency of the pump prototype: (**a**) photograph of the experimental site and (**b**) flow rate versus frequency at 150 V_p-p_.

**Figure 9 micromachines-14-01733-f009:**
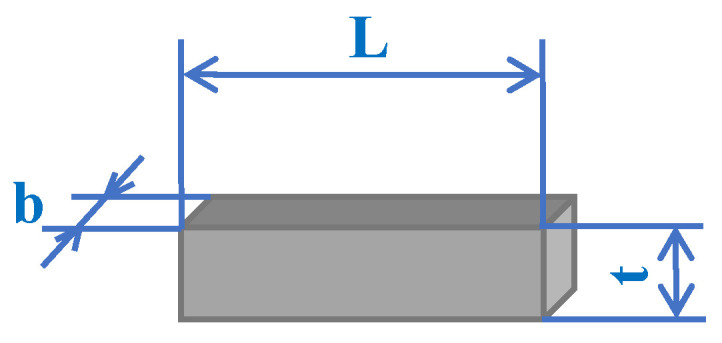
Simplified model of the central junction in the cantilever valve.

**Figure 10 micromachines-14-01733-f010:**
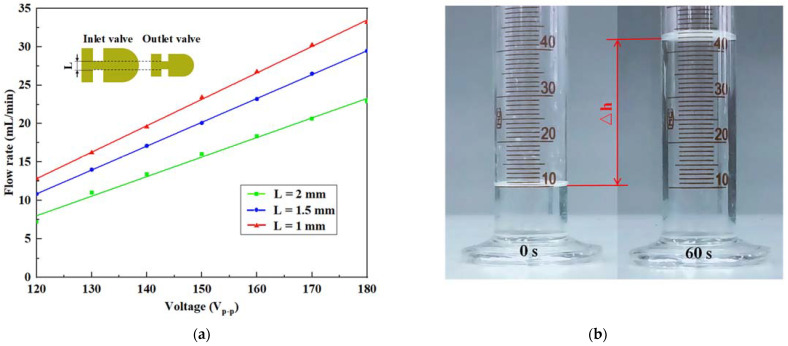
Optimization experiments of the valve system: (**a**) flow rate versus voltage under different stiffness valves and (**b**) flow rate change of the valve system with *L* of 1 mm at 180 V_p-p_.

**Figure 11 micromachines-14-01733-f011:**
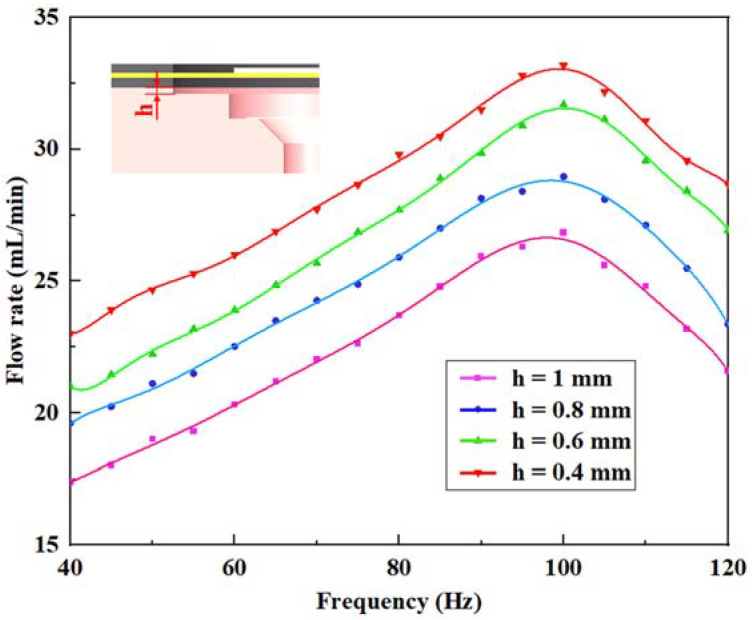
Flow rate versus frequency under different heights of the pump chamber.

**Figure 12 micromachines-14-01733-f012:**
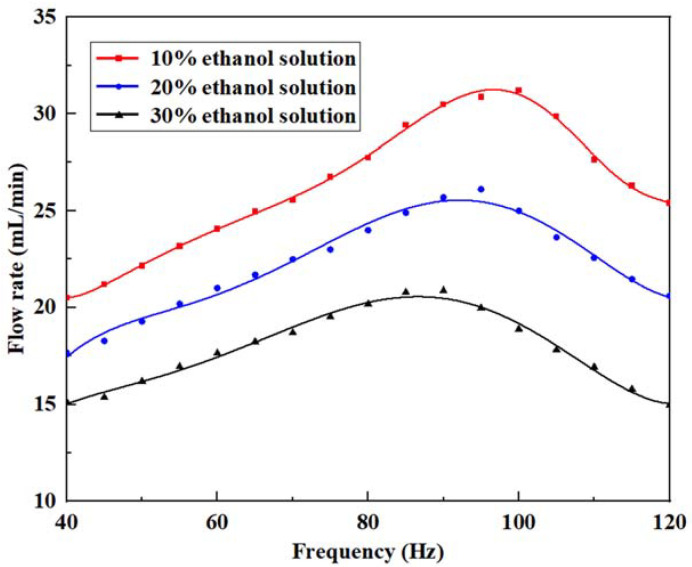
Flow rate versus frequency under different viscosities of ethanol solution.

**Table 1 micromachines-14-01733-t001:** Geometrical parameters of the upper pump body structure (unit: mm).

Parameter	R_1_	R_2_	d_1_	d_2_	Φ_1_	Φ_2_	H	h_1_	h_2_	h_3_	s_1_	s_2_
Value	45	25	5	3	35	30	4.5	1.2	0.1	0.8	6.5	5.5

**Table 2 micromachines-14-01733-t002:** Material properties of the piezoelectric actuator.

Material Properties	PZT-5H			Brass	Water
Density (kg/m^3^)	7500			8800	1000
Poisson’s ratio	0.31			0.324	x
Elastic modulus (×10^10^ N/m^2^)	12.155.355.15000 12.065.15000 10.45000 3.1300 3.130 3.46			x	x
Piezoelectric constant (C/m^2^)		$\left[\begin{array}{ccc} 0 & 0 & -5.2\\ 0 & 0 & -5.2\\ 0 & 0 & 15.1\\ 0 & 12.7 & 0\\ 12.7 & 0 & 0\\ 0 & 0 & 0 \end{array}\right]$		x	x
Speed of sound (m/s)	x			x	1482.1

**Table 3 micromachines-14-01733-t003:** Comparison of various cantilever valve piezoelectric pumps.

Authors	Output Flow (mL/min)	Frequency (Hz)	Voltage (V_p-p_)
Kan et al. [36]	3.5	200	50
Ma et al. [38]	2.35	130	50
Woo et al. [39]	18.7	110	100
Liu et al. [40]	25.2	120	100
Yang et al. [41]	4.5	150	180
This work	33.18	100	180

## Data Availability

No new data were created or analyzed in this study. Data sharing is not applicable to this article.

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
