# Peer review of "A Dual-Inlet Pump with a Simple Valves System"

_micromachines, 2023, doi:10.3390/mi14091733_

Round 1
Reviewer 1 Report
In this paper, in order to improve the output flow of the cantilever valve piezoelectric pump is small, a double inlet piezoelectric pump is proposed, but there are still some problems that need to be improved, the specific problems are as follows:
1. The number of bolt holes in the top view in Figure 2 is different from the actual one and needs to be modified.
2. In chapter 4.3, the size parameter of the valve system is expressed as 1 mm L, which is not standard and needs to be modified.
3. In chapter 4.3, the effort of L on the stiffness of the cantilever valve should be correctly described to illustrate the effect of valve stiffness on the output flow rate.
4. Figure 9 shows the effect of valve stiffness and output voltage on the output flow of a piezoelectric pump. The input voltage is positively correlated with the output flow, why not test the output flow at 190 Vp-p and 200 Vp-p, where a higher flow is possible.
5. The design objective of this paper is to improve the output flow of the cantilever valve piezoelectric pump, and it is necessary to compare the performance with other piezoelectric pumps to prove the scientific of the design.
Have no opinion
Author Response
Reviewer: 1
In this paper, in order to improve the output flow of the cantilever valve piezoelectric pump is small, a double inlet piezoelectric pump is proposed, but there are still some problems that need to be improved, the specific problems are as follows:
Response: Thanks a lot for your suggestions for this work! We have rechecked our paper and found that your suggestions are really of great value for improving it. Here the problems have been revised one by one. We hope these changes can enrich our article content and make it more readable.
Comment 1: The number of bolt holes in the top view in Figure 2 is different from the actual one and needs to be modified.
Response: Thank you very much for your comment. We are sorry for the wrong picture selection. Now Figure 2 has been corrected.
Comment 2: In chapter 4.3, the size parameter of the valve system is expressed as 1 mm L, which is not standard and needs to be modified.
Response: Thank you very much for your comment. The expression of L has been adjusted in chapter 4.3.
Comment 3: In chapter 4.3, the effort of L on the stiffness of the cantilever valve should be correctly described to illustrate the effect of valve stiffness on the output flow rate.
Response: Thank you very much for your comment. We take the central connecting part of a single valve plate and simplify it into a rectangular model to obtain its equivalent stiffness expression as Eq. 6. Due to the positive correlation between stiffness and L, reducing the size of L helps to reduce stiffness. These discussions are reflected in the relevant section on Page 9.
Comment 4: Figure 9 shows the effect of valve stiffness and output voltage on the output flow of a piezoelectric pump. The input voltage is positively correlated with the output flow, why not test the output flow at 190 Vp-p and 200 Vp-p, where a higher flow is possible.
Response: Thank you very much for your question. During the experiment, we observed that 180 Vp-p is the critical value of breakdown voltage. When higher voltage such as 190 Vp-p or 200 Vp-p is input, the piezoelectric actuator will usually be damaged. Therefore, 180 Vp-p is selected as the upper limit of the experimental voltage to ensure the safety of the prototype and obtain obvious experimental results. This point has been briefly expressed in Line 276 and 277 on Page 10.
Comment 5: The design objective of this paper is to improve the output flow of the cantilever valve piezoelectric pump, and it is necessary to compare the performance with other piezoelectric pumps to prove the scientific of the design.
Response: Thanks you very much for your precious advice! We selected some typical piezoelectric pump results using cantilever valves in Table 3 on Page 11. And then compared them with our proposed pump for flow rate, frequency, and voltage. Maybe it can make the features of our manuscript more prominent and persuasive.
Thanks again for your comments above!

Reviewer 2 Report
Present manuscript by Wang et al., discusses about the performance ofa piezoelectric pump with valve attached to the pump. Article is well presented and can be published. However, I have the following concern(s) which the authors may consider before a final revision.
(1) Looks like commercial piezoactuator has electrode on both side. Under this configuration, piezoelectric shows linear displacement not buckling. However, working mechanism shown here entertains the buckling of actuator under the field application. Can authors explain about this discrepancy ?
(2) Authors need to explain the reason behind decrease in flow rate beyond certain frequency?
(3) Why authors considered a disk shaped actuator. A cantiliver type actuator might be appropriate.
(4) Does the pumping performance depends on the nature of the liquid? Authors need to discuss about it.
Author Response
Reviewer: 2
Present manuscript by Wang et al., discusses about the performance of a piezoelectric pump with valve attached to the pump. Article is well presented and can be published. However, I have the following concern(s) which the authors may consider before a final revision.
Response: Sincerely thanks for your recognition and suggestions! Several points you have raised are very meaningful for improving and enriching our manuscript. We have answered these comments and made some supplements in the article. Although our current work still has many deficiencies, we will make further efforts in the following researches.
Comment 1: Looks like commercial piezoactuator has electrode on both side. Under this configuration, piezoelectric shows linear displacement not buckling. However, working mechanism shown here entertains the buckling of actuator under the field application. Can authors explain about this discrepancy ?
Response: Thank you very much for your question. For the piezoelectric actuator used in this work, the positive electrode is on the surface of the PZT transducer, and the negative electrode is on the side where the metal substrate is bonded to the PZT. Not all sides have electrodes. We choose the first order mode, which is the thickness direction stretching vibration mode, as the operating mode. The displacement of particles on the surface of piezoelectric actuator is linear in the out of plane direction, it reaches the largest value at the center and shows a gradually decreasing trend along the radius direction (it can seen in simulation and vibration test results). Therefore, the piezoelectric actuator appears a strain effect of reciprocating bending up and down.
Comment 2: Authors need to explain the reason behind decrease in flow rate beyond certain frequency?
Response: Thank you very much for your comment. The 100 Hz frequency observed in the experiment is correspond with the simulated resonant frequency of the piezoelectric actuator. The amplitude peak appears at this point, and the pump output flow performance is optimal. Then the amplitude decreases (as shown in Figure 5 and Figure 6) after exceeding 100 Hz, which makes the flow rate goes down. The explanation for this phenomenon has been added on Page 9, Line 255 to 258.
Comment 3: Why authors considered a disk shaped actuator. A cantiliver type actuator might be appropriate.
Response: Thank you very much for your question. Firstly, the cantilever piezoelectric actuator is adverse to fixing with the pump body structure. Secondly, its asymmetric strain will cause uneven changes in the pump chamber volume, which may result in insufficient continuous and stable output flow. So this type is not conducive to experimental observation and the improvement of pump flow. On the contrary, the disc shaped piezoelectric actuator can be symmetrically fixed to the pump body. From the cross-sectional perspective (see Figure 3), the change in pump chamber volume is also symmetrical, so the liquid can be continuously and uniformly sucked in and discharged.
Comment 4: Does the pumping performance depends on the nature of the liquid? Authors need to discuss about it.
Response: Thank you very much for your suggestion. The nature of the liquid itself has an impact on the flow output of the pump. Here we further investigated the effect of viscosity, an inherent property of liquids, on the flow rate of pump. We selected ethanol solutions of different concentrations for exploration, and the experimental results and phenomenon analysis are presented in Chapter 4.5 on Page 11 to 12. We hope that adding this section can enrich the content of our manuscript.
Thanks again for your comments above!
